# Skin Immuno-CometChip in 3D vs. 2D Cultures to Screen Topical Toxins and Skin-Specific Cytochrome Inducers

**DOI:** 10.3390/genes14030630

**Published:** 2023-03-02

**Authors:** Dean S. Rosenthal, Li-Wei Kuo, Sarah L. Seagrave, Vikas Soni, Nusrat Islam, Geetanjali Minsky, Lucia Dussan-Cuellar, Brian Ell, Cynthia M. Simbulan-Rosenthal, Peter Sykora

**Affiliations:** 1Department of Biochemistry and Molecular & Cellular Biology, Georgetown University School of Medicine, Washington, DC 20057, USA; 2Amelia Technologies, LLC, Washington, DC 20001, USA

**Keywords:** genotoxicity, skin equivalents, integrin β1, cytochrome p450

## Abstract

The targets of topical genotoxic agents are basal and stem cells of the skin. These cells may misrepair DNA lesions, resulting in deleterious mutations of tumor suppressors or oncogenes. However, the genotoxicity of many compounds has not as yet been determined and needs to be tested using a relevant skin model. To this end, we designed a new high-throughput assay for the detection of agents that create DNA damage in epidermal stem and basal cells and used it to test known DNA-damaging agents. We utilized either 2D epidermal cells or 3D skin equivalents and topically exposed them to different compounds. The Skin Immuno-CometChip assay uses arrays of microwells formed in a collagen/agarose mixture to capture single basal cells in each microwell by virtue of collagen binding to α2β1 integrin, which is present only on basal and stem cells. The presence of β1 integrin was verified by immunofluorescent labeling cells that were then subjected to an electrical field, allowing for the migration of nicked DNA out of the nucleoid in alkali, with the resulting DNA comets stained and imaged. Furthermore, using improved comet detection software allowed for the automated and rapid quantification of DNA damage. Our study indicates that we can accurately predict genotoxicity by using 3D skin cultures, as well as keratinocytes grown in 2D monolayers.

## 1. Introduction

The skin is the largest organ and the first to come into contact with environmental carcinogens, either incidentally, such as through exposure to household chemicals, or via the application of topical agents such as cosmetics and sunscreens. In spite of the best efforts of regulatory agencies to ensure consumer safety, the sheer volume of chemicals in everyday products makes it impractical to screen them effectively by using low-throughput models. Roughly 12,500 cosmetics are on the US market at any given time [1]. Furthermore, at a replacement rate of roughly 25–30%, 3000 to 4000 cosmetics need to be tested each year. Most of the 85,000 chemicals commonly used in the US, outside of drugs and pesticides, and including cosmetics, personal products, household cleaners, food, fabric, and children’s toys, have not been tested for their effects on human health. More broadly, 10 million chemicals are currently being produced each year with little or no analysis of health or environmental impact [2]. Where there has been any testing at all, it has been historically performed on animals, in conjunction with other assays. These have been used to generate databases, including those of the National Toxicology Program (NTP) (available online: https://ntp.niehs.nih.gov/data/index.html?utm_source=direct&utm_medium=prod&utm_campaign=ntpgolinks&utm_term=datasearch (accessed on 27 February 2023)) and the International Agency for Research on Cancer (IARC; available online: https://publications.iarc.fr/Databases/Iarc-Cancerbases (accessed on 27 February 2023)). However, animal testing has been phased out in Europe and has recently been prohibited in some US states. Even assuming that most of these chemicals are safe, overlooking certain chemicals because of lack of testing or political will has had major public health and ecological consequences, as in the cases of arsenic, asbestos, benzene, bisphenol A (BPA), chromium hexavalent compounds, dioxins, formaldehyde, polybrominated diphenyl ethers (PBDEs), polycyclic aromatic hydrocarbons (PAH), and vinyl chloride. A number of these chemical agents are both genotoxic and carcinogenic [3]. Accumulated DNA damage and misrepair results in mutations, most of which are inconsequential. However, damage can also result in the accrual of deleterious mutations and ultimately tumorigenesis when oncogenes or tumor suppressor genes are altered [4]. Thus, the rationale for the well-known Ames mutation test is precisely due to the fact that a large proportion of genotoxic agents are also carcinogens. Another screening challenge is that once a single compound has been approved, it no longer needs to be tested in combination with other compounds, which may act synergistically with each other to generate genotoxic combinations.

There are several established methods for measuring DNA damage and resulting mutations. Since rodent models are not practical, in vitro genotoxicity can be measured using several different techniques, including the aforementioned Ames test, mouse lymphoma assay, and in vitro micronucleus and chromosomal aberrations test. Higher sensitivities could be achieved using combinations of these assays [5]. The prokaryote-based Ames test employs a Salmonella typhimurium reverse mutation assay (*his*− to *his*+) [6]. The mouse lymphoma assay similarly measures gene mutation, usually TK, in eukaryotes [7]. Some of these assays can be time-consuming and can yield a high proportion of false-positive results [5]. In contrast, single-cell gel electrophoresis (SCGE), or the Comet, is now internationally recognized as a simple and inexpensive method for the detection of different types of DNA damage, including single- and double-stranded breaks, DNA adducts, crosslinks, and alkaline-labile sites, and has been used in a low-throughput format until recently [8,9,10,11,12,13,14,15,16,17,18].

While DNA damage can occur in any organ, the skin withstands particularly relentless assaults from the environment and has evolved several mechanisms to protect its basal layer from DNA damage and mutation. The outer anucleate cornified skin layers serve as a first line of defense, protecting the proliferating basal cells that are resident in the lowest layer of the epidermis; protection is paramount, as mutations in stem and basal cells are necessary for tumorigenesis. Furthermore, UVA (320–400 nm) and some genotoxic chemicals can penetrate to the dermis, while other chemicals may leach into the bloodstream, as in the case of sunscreen active ingredients such as oxybenzone (clinical trial NCT03582215) [19]. Additionally, the skin contains xenobiotic-metabolizing enzymes, including three type I cytochrome P450s (**CYPs**), namely CYP1A1, 1A2, and 1B1, which oxidize carcinogens, forming intermediates that can react with DNA to form mutagenic adducts [20]. Thus, it is important to have a reliable high-throughput method to determine the skin genotoxicity of chemicals that are likely to come into contact with humans. The need for data that can be extrapolated to humans has generated interest in organotypic human skin models that recapitulate biochemical and morphological properties of human skin, including those related to absorption and drug metabolism. Numerous endpoints have been validated and adopted, including corrosion, irritation, and sensitizing assays. Currently, there are no standardized assays to assess genotoxicity in the skin. Part of the difficulty lies in the nature of skin, which, like lens fiber and reticulocytes, contains anucleate terminally differentiated cells. Thus, while different assays are available for the screening of DNA breaks, these methods are complicated in mature skin, since the degradation of DNA is part of the normal process of epidermal differentiation, resulting in a well-organized stratified squamous epithelium, enabling its barrier function against environmental insult. It is in this context of differentiated epidermis that environmental carcinogens act on the skin. Thus, there has been growing interest in developing a reliable skin comet assay.

We combined the advantages of CometChip with those of organotypic culture. The ability to grow human basal-like cells in specialized low calcium medium containing growth factors and specific concentrations of retinoids allows for the investigation of both the short- and long-term effects of DNA damage on keratinocytes. However, the response of monolayer cells does not replicate the in vivo response to toxic insult. On the other hand, organotypic and/or xenograft skin cultures on immune-compromised mice have successfully recapitulated the response of the skin to toxic agents and irradiation, including signature cytokine, metabolomic, and transcriptomic profiles [21]. Among other factors, these differences may be explained by differences in p21Cip1 cyclin-dependent kinase inhibitor levels inducing apoptosis in monolayers, as well as altered differentiation in the organotypic culture, with “bystander” cells contributing to the latter effect [21]. However, this creates a conundrum: under normal growth conditions, monolayer cells are “basal-like” but lose their in vivo response to toxins, while organotypic cultures, such as human epidermis, lose their nuclei during differentiation, creating a heterogeneous background of DNA damage in published reports of skin comets [22]. Secondly, it has long been known that proliferating basal cells have a more robust DNA-repair response than differentiated keratinocytes derived from the same epidermis [23]. The third problem is the identification of the cell type in the skin that has sustained damage, as full-thickness skin cultures contain both fibroblasts and keratinocytes.

This platform thus adds three new parameters to the Comet assay: higher throughput, cell-type isolation and identification, and increased accuracy for DNA-damage measurements. The uniqueness of this model is thus the use of an organotypic human skin model that it more accurately recapitulates the response of human skin than other cell models of genotoxicity currently used for screening and the ability to isolate basal cells, the target of genotoxic and carcinogenic agents, and precisely and rapidly measure their DNA damage in a high-throughput CometChip format.

## 2. Materials and Methods

### 2.1. Cells

E6/E7-immortalized or HaCaT keratinocytes were maintained in DMEM media supplemented with 10% FBS and 1% of Penicillin–Streptomycin (ThermoFisher, Waltham, MA, USA, 10,000 U/mL). Conditionally reprogrammed keratinocytes (CRC-HFK) were maintained in EpiX medium according to the manufacturer’s protocol (Propagenix, Gaithersburg, MD, USA). Human Foreskin Keratinocytes (HFKs) were kind gifts from Richard Schlegel at Georgetown University Medical Center and were co-cultured in FY medium (3 parts of DMEM containing 10% FBS, 1% of Penicillin–Streptomycin, 1% of 100X glutamine, and 1 part of F12 nutrient mixture) plus Hydrocortisone/EGF (1:1000), insulin (4 µg/mL), Gentamicin (10 μg/mL), Fungizone (250 ng/mL), 0.1 nM cholera toxin, and 10 μM ROCK Inhibitor Y-27632. Jurkat T cells were cultured in RPMI supplemented with 10% FBS and 1% of Penicillin–Streptomycin. All cells were cultured at 37 °C in a 5% CO_2_ cell incubator.

### 2.2. Skin Organotypic Cultures

Full-thickness skin equivalents (EFT-400, MatTek, Ashland, MA, USA) and epidermal equivalents (EPI-201) were purchased from MatTek in 24-well formats, while EpiX-HFK skin equivalents were generated in our lab. MatTek tissue was maintained in DMEM for 3 days prior to exposure to compounds. Dissociated skin equivalents or detached monolayer cells were blocked with 6% BSA in PBS for 30 min and then hybridized with biotin-conjugated anti-human integrin β1 (1:200 Miltenyi Biotec cat# 130-101-262) for 1 h at room temperature. After 3 PBS washes, cells were hybridized with Qdot™ 655 Streptavidin Conjugate (20 nM, Invitrogen cat# Q10121MP) for 1 h. Cells were washed with PBS 3 times and resuspended in cell culture media prior to loading into the CometChip.

### 2.3. Immunostaining

For chamber slide staining, 30 K cells were loaded into each chamber; after attachment, cells were washed and fixed with 4% paraformaldehyde (PFA) for 15 min, permeabilized with 0.5% Triton X-100 (in PBS) for 30 min, and then hybridized with anti-human integrin β11 antibody and fluorescein-conjugated phalloidin (1:40, F432, ThermoFisher Scientific) overnight at 4 °C. The following day, cells were PBS-washed 3X, and incubated with Qdot™ 655 Streptavidin Conjugate (20 nM) for 1 h, along with DAPI (1:2000, D1306, Invitrogen) to visualize the nucleus. For the 3D skin tissue stain, after treatment with compounds, the membranes in culture inserts were removed, paraffin-embedded, and sectioned according to standard protocols by Histopathology & Tissue Shared Resource at Georgetown University Medical Center. For immunofluorescent staining, sections were deparaffinized in xylene for 5 min 2 times and then rehydrated in 100%, 90%, 70%, and finally 30% ethanol for 1 min each. Then antigen retrieval was performed using 10 cycles of microwave boiling and cooling. After a brief PBS wash, tissues were permeabilized with 0.2% glycine/PBS for 15 min and then blocked with SuperBlock^TM^ Buffer (ThermoFisher cat #37515) for 1 h. Tissues were incubated overnight at 4 °C, using CYP1A1 (Abcam, Cambridge, UK, Ab3568, 1:1000) or rabbit Anti-CYP1B1 antibody Ab33586 1:1000). The next day, after three PBS washes, the tissue was hybridized with corresponding secondary antibody at room temperature for 1 h: Alexa-488-conjugated goat anti-mouse IgG - (Abcam Ab150113, 1:500) or Goat anti-rabbit IgG -Alexa 594 conjugated (Invitrogen # A32740, 1:500). DAPI (Invitrogen, Waltham, MA, USA) stain was used to visualize the nucleus.

### 2.4. Treatment of Cells with Compounds

A total of 50K cells/well were loaded into the comet chips. After 30 min of gravity loading, as previously described [24], cells were treated with NTPs compounds or known DNA-damage agents at indicated concentration for 1 or 3 h. DMSO was used as a negative control.

### 2.5. Cell Viability Assays

First, 10^4^ cells were seeded into 96-well culture plates. After 16 h to allow attachment, the cells were treated with 200 µM of each compound and exposed for 3 h at 37 °C in a 5% CO_2_ cell incubator. Compounds were then removed, the cells were washed with PBS, and the original culture medium was added back. Measurements were conducted 24 h after exposure, using an XTT Cell Viability Assay Kit (Biotium, Fremont, CA, USA) according to the manufacturer’s instructions, using Absorbance 450- Absorbance 650 at 0, 1, 2, and 3 h. To calculate % cell viability, the absorbance/time slopes were determined from four time points, as mentioned, and % cell viability = (the slope of samples/the slope of DMSO- or untreated).

### 2.6. CometChip Assay and Analysis

CometChips were cast using a mixture of 1% agarose and 1 mg/mL collagen (Gibco, Bovine Collagen I); all other physical characteristics remain as previously described [24]. The collagen added to the CometChip allows for the binding of only β1-integrin-expressing basal keratinocytes. The comet chip assay system (R&D system) was used according to manufacturer’s instructions. In our previous study, microwells that were 30 µM diameter were able to trap most of the cells. Gels comprised agarose (0.8% w/v) or agarose containing 1 mg/mL of collagen I (Gibco, New York, NY, USA, A1048301), as follows: First, 1.6% NA (normal agarose (Invitrogen, #16500-100) was dissolved in 100 mL 1X PBS pH 7.4 (Invitrogen, 10010-049) in a 250 sterile flask, using a microwave, and adding sterile water until the original mass is restored. The temperature of the gel was then equilibrated to 55 °C in a water bath before mixing with 2 mg/mL collagen I. For long-term storage, 30 mL aliquots can be stored in 50 mL conical tubes at 4 °C. A 2 mg/mL collagen I solution was prepared separately in a 10 mL conical tube. Then 4 mL of sterile dH_2_O was added to 1 mL 10X PBS. The mixture was heated to 90 °C, and then 4 mL of 5 mg/mL rat-tail collagen I (Gibco, cat# 1048301) was added and mixed by inversion. The solution was cooled to room temperature, and the pH was adjusted to 7.4, using fresh 7.5% Sodium Bicarbonate (60–240 µL, verified by a pH meter). The temperature was equilibrated to 55 °C before mixing with the 1.6% agarose solution. For long-term storage, the 2 mg/mL collagen I can be stored at 4 °C. In a 50 mL conical tube, 7 mL of 2 mg/mL collagen I (equilibrated to 55 °C) was added to 7 mL 1.6% agarose (equilibrated to 55 °C) and poured slowly into the CometChip mold. The gel was allowed to harden for 30 min at room temperature. Cells were counted to achieve 5 × 10^5^/mL, and then 100 µL (50K cells) was loaded into each of the 96 macro-wells and trapped by gravitation and integrin–collagen interactions. After 30 min, the cells were exposed to the indicated chemical compounds or positive control DNA damaging agents. All incubations were performed at 37 °C, 5% CO_2_. We analyzed comets by using Comet Analysis Software, as described below.

The CometNet software initially makes a grayscale copy of the macro-well stitched image. This allows for the measurement of pixel intensities required for downstream analysis. Next, CometNet does a primary scan of the image with a fixed threshold value to separate the foreground from the background. It then utilizes the “findContours” function from the OpenCV module (Open Source Computer Vision Library; http://opencv.org (accessed on 27 February 2023) to find all the contours in the image; these are defined as “comet candidates” or potential comets. It further filters the comet candidates via a “for loop” with specified parameters that indicate a “true comet”; parameters include a bounding box that is either square for comets with low damage or rectangular for comets with damage above ~10% damage. Once it finds the candidate comets, the program extracts the bounding box and saves a sub-image of each comet. On the extracted bounding box sub-image, the CometNet performs a further analysis to find the comet head and calculate the key statistics for the individual comet damage profile. The CAS interrogates the selected bounding box for each comet and takes a square region on the leftmost side of the bounding box where the head will be in all comets. This method improves upon the previous CAS ability to find heads that appear detached from comet tails, as the square bounding box that CometNet draws will always include the full head. From here, the program utilizes image morphology techniques, including erosions and dilations and the findContours () function to include the corona shape of the head. It then draws a contour of this corona and fills it in to create a “mask”, which specifies the head for further calculations. The program has a cometStats () function which outputs the width, height, total comet area, and tail area of the comet into a CSV file. The area is the sum of pixel intensities in a contour. The area and intensity of the DNA in the tail is correlative with the amount of DNA damage in the cell.

### 2.7. qRT-PCR (Quantitative Reverse-Transcriptase-Mediated PCR)

Total RNA was isolated using the RNeasy mini Kit (74104, Qiagen, Venlo, The Netherlands). A total of 1 µg of RNA was converted to cDNA by using the Verso cDNA Synthesis Kit (AB1453A, ThermoFisher Scientific). The qPCR mixture contained the cDNA diluted 50-fold in nuclease-free water.

The primer pairs Human CYP1A1- F: CAAGAGGAGCTAGACACAGTGATT; Human CYP1A1- R: AGCCTTTCAAACTTGTGTCTCTTGT; Human CYP1B1- F: TTCGGCCACTACTCGGAGC; Human CYP1B1- R: AAGAAGTTGCGCATCATGCTG; Human CYP1A2- F: TGGCCTCTGCCATCTTCTG; and Human CYP1A2- R: GGACCCGAGGCCTCAAAC and SYBR Green Master Mix (Applied Biosystems, Waltham, MA, USA) were added, and the signal was detected by a MiniOpticon Real-Time PCR System (BioRad, Hercules, CA, USA). GAPDH served as an internal control. The fold-change was calculated based on the formula 2^−△△CT^, where △△C_t_ = [C_t_ (CYP)_TREATMENT_ − C_t_ (GAPDH)_TREATMENT_] – [C_t_ (CYP)_CONTROL_− C_t_ (GAPDH)_CONTROL_]. C_t_ = threshold response; CONTROL = solvent alone.

### 2.8. CYP450 Activity Assays

CYP1A1/CYP1B1 activity was measured using Luciferin-CEE as a substrate, according to the manufacturer’s instructions (P450 Glo Assay, Promega, Madison, WI, USA). Luminescence was detected using an Enspire® Alpha Plate Reader (PerkinElmer, Waltham, MA, USA) and steady-state luminescence plotted. Human S9 Fractions (Catalog number HMS9PL; ThermoFisher) were purchased at a stock concentration of 20 mg/mL and diluted as controls for the CYP1A1/1B1 assay.

### 2.9. Statistical Analysis

All experiments were performed in at least duplicate, with representative experiments shown. One-way ANOVA was used, with *p*-values < 0.05, 0.01, 0.001, and 0.0001 considered significant and designated with one, two, three, or four asterisks, respectively.

## 3. Results

### 3.1. Development of Control Cells with Fixed Amounts of DNA Damage

In the original CometChip description, we included one positive and one negative DNA damage control [24] in each assay run. The DNA damage in the positive control was induced using the clinical topoisomerase inhibitor, etoposide (Sigma-Aldrich, St. Louis, MO, USA), which creates single- and double-stranded breaks that are measurable by single-cell gel electrophoresis. In the Immuno-CometChip iteration described herein, the onboard controls were expanded to include a concentration curve of DNA damage induced by exposing HaCaT human keratinocyte to etoposide (Appendix A). The shift from a single DNA damage positive control per assay to a DNA damage control curve was vital to be able to compare data more accurately from multiple CometChip runs. DNA damage concentrations of etoposide were determined to be linear, between 0 and 10 μM (Appendix A), with higher concentrations failing to produce a linear response (Appendix A).

The second objective of these experiments was to create a bank of frozen control DNA damage samples. To this end, a storage protocol was previously described for reptile nucleated blood cells [24]. Here, we adapted the protocol for mammalian cells (see Materials and Methods). The cells were thawed after freezing intervals ranging from one day to one month, and DNA-damage levels measured (Appendix A). The freezing protocol increased the measured DNA damage after four weeks of storage (green data points, Appendix A); however, a concomitant increase in freezing-associated DNA damage levels was not seen in the treated sets after the same period (red data points, Appendix A). The full DNA-damage curve was also tested for stability after three months of low-temperature storage (Appendix A). The DNA-damage levels remained stable after the thawing of the samples, with a stable linear increase in the amount of DNA damage measured.

### 3.2. Development of Next-Generation Computer Software to Differentiate Integrin β1-Positive Cells

#### 3.2.1. Redevelopment of the Comet Measurement Analytics

Comet analysis software (**CAS**) was developed to identify and count SCGE comets [24]. However, the software had shortcomings that required a new CAS code to be developed. Specifically, in high-damage cells, we found that the tail was often mis-identified by the existing CAS as the comet head (Figure 1A). A newly developed CAS (“CometNet”) was optimized for higher-precision comet finding and detection using integrated machine learning. CometNet differs from other previous CASs because it possesses the following attributes: it is written in Python and OpenCV, a powerful scientific language, and uses a heuristic approach to locate the comet head. CometNet uploads unaltered images of individual CometChip macro-wells taken with an automated cytometer (Cytation5, Biotek), with each macro-well containing between 300 and 500 comets enclosed in individual micro-wells [24].

#### 3.2.2. Comparison of CometNet to Commercial CAS and Freeware CAS

To determine the effectiveness of the new CAS, we compared it to a commercial CAS (developed by Trevigen, currently sold by Biotechne, Minneapolis, MN, USA) and a free-ware CAS (OpenComet, https://cometbio.org/ (accessed on 19 December 2022)). HaCaT keratinocytes were irradiated using UVA (365 nm) from a proprietary high-throughput irradiation system. The images were then analyzed using the three aforementioned CASs. We measured variables such as image processing time, total comets found, erroneous comets found, and average DNA damage detected in each well. The three CASs were tested with image data from five different fluence exposures (“doses”) of UVR that produced linearly increasing amounts of DNA damage (0–142 kJ/m^2^; UVA 365 nm; Figure 1A). All CASs were able to analyze the images with comparable linear regression values (CometNet, r^2^ = 0.96; Trevigen, r^2^ = 0.97; and OpenComet, r^2^ = 0.98). However, the different algorithms for analyzing the images did give different calculations of DNA-damage levels, particularly when comparing Trevigen CAS and OpenComet. These two CAS programs differed significantly from each other (*p* < 0.05) at every exposure tested. As depicted in Figure 1A, the difference between the two CASs remained constant (approximately 20% DNA tail) at each data point. The ability of a CAS to correctly identify all comets in an image is also paramount. To test this, 27 wells were loaded with cells having either low (<10%), moderate (10–45%), or high levels (>45%) of DNA damage (Figure 1B). When they were inputted with images from cells with low levels of DNA damage, all the three CASs did comparably well. Overall, CometNet was able to significantly identify (*p* < 0.0001) more cells at all DNA-damage levels, with the Trevigen CAS consistently identifying the fewest cells per macro-well.

### 3.3. Validation of the Basal Cell Surface Marker Identification System

Since the most consequential target of epidermal DNA damage is stem and basal cells, we set out to determine if we could identify and enrich this population in our assays. We therefore used β1-integrin expression for basal cell identification since high levels have been shown to correlate with a stem/basal cell phenotype [25]. β1-integrin expression in HaCaT, normal human epidermal keratinocytes (NHEKs), and conditionally reprogrammed CRC-HFK was confirmed using immunocytochemistry (Figure 2). CRC-HFK, NHEK, HaCaT, and Jurkat T cells were fixed and stained with biotinylated integrin-β1 antibody subsequently coupled to Qdot 655. The cytoskeleton compartment was co-labeled with FITC-conjugated anti-F-actin (Figure 2). All keratinocyte lines showed integrin-β1–Qdot labeling. In contrast, the Jurkat lymphocytes did not support Qdot labeling.

To determine our ability to isolate and label basal keratinocytes in the Immuno-CometChip Assay, collagen I (1 mg/mL) was incorporated into low-melting-point agarose gel, and then microwell-stamp-generated collagen plus low-melting-point agarose composite gel was cast to which basal keratinocytes selectively bound via integrin β1 on the basal cell membrane. Two important variables to select for basal cell binding to collagen include the size (diameter and depth) of the microwells, and the rigor and number of washing steps after cells are loaded into the wells. The microposts used for these microwells [26] have a variable diameter that can be altered from 10 to 50 μm. While 30 µm wells can accommodate most cell types, other sizes were tested to determine optimal collagen/integrin binding. Washing steps were then performed to remove non-basal cells, which lack basal integrins. We then altered different parameters, such as microwell sizes, to test the optimal binding of basal keratinocytes to collagen.

To show selective attachment of basal keratinocytes to microwells, Jurkat (β1 integrin-low T cells) and HaCaT keratinocytes (β1-integrin-high basal epidermal keratinocytes) were used as negative and positive controls, respectively. To remove cell surface receptors, including β1-integrin, a portion of the HaCaT cells were over-digested with trypsin for 10 min. Agarose +/− collagen CometChips containing 20, 30, and 40 µ wells were then prepared. A total of 30,000 cells were loaded per CometChip macro-well and incubated for 30 min in a CO_2_ incubator. The CometChip was gently rinsed with PBS, and 4X well images were taken (Figure 3A), followed by a second rinse with PBS, with 4X well images taken again. An analysis of the cell number for each well image was determined using Trevigen Comet Analysis Software (graphs, Figure 3B,C). After one PBS wash, both β1 integrin-negative Jurkat and β1-integrin-positive HaCaT keratinocytes remained in the microwells with diameters of 30–40 μm and a depth of 50 μm (Figure 3B). However, after two PBS washes, integrin-β1-low Jurkat cells, as well as HaCaT cells that were stripped of integrin β1 by over-trypsinization, were washed out of the 30 μm wells. In contrast, integrin-β1-positive HaCaT cells remained in the wells since the integrin heterodimers containing β1 interact and bind strongly to the collagen I in the gel (Figure 3C).

For our first 3D skin experiment, we applied H_2_O_2_ to the stratum corneum of the EpiDerm (MatTek, Ashland, MA, USA) prior to its dissociation. EpiDerm cultures of 9 mm in diameter (24 wells; 0.63 cm^2^ each, containing 0.7 to 1.5 million basal cells; [27]) were dissociated with trypsin for 5 min (4 °C); the cells were pelleted, PBS washed, and then re-pelleted. During this time, cells were kept at 4 °C during the assay to limit any unintended DNA damage. We were able to easily visualize basal cells derived from EpiDerm. While Jurkat cells were washed out of the wells of 30 µ in diameter (Appendix A, left panel), EpiDerm basal keratinocytes remained in the wells after two PBS washes, presumably because of the interaction of collagen I with α2β1 integrin on the basal cell membrane. To determine if this was in fact the case, after exposure of EpiDerm to H_2_O_2_, dissociation, microwell plating, and washing, the cells remaining in the microwells were incubated with biotinylated anti-integrin β1, followed by Streptavidin Qdot 655. All EpiDerm cells in the wells were positive for integrin β1 and were therefore stem/basal keratinocytes (Appendix A middle). Cells were stained with SYBR Gold before electrophoresis (Appendix A, right) to visualize DNA, which, like Qdots, remained in the head of the comet. EpiDerm 3D cultures were next exposed to increasing doses of H_2_O_2_, then dissociated, and cells were again plated in microwells and stained for β1, then subjected to CometChip electrophoresis, followed by DNA staining with SYBR Gold. Qdots remained in the well at the head of the comet (yellow circles), while the DNA tail was clearly visible as a green streak (Appendix A). The dose response to H_2_O_2_ was robust (Appendix A). This system is therefore a powerful tool for (1) isolating basal cells, (2) determining their level of DNA damage, and (3) validating the basal cell origin of comets prior to including them in the measure of DNA damage, as well as confirming the compatibility and ease of use of multiple markers.

### 3.4. The Integration of Cell Location Registration into the Comet Analysis Software

There are limitations to the use of the antibody-Qdot-labeling approach. Foremost is that, after organotypic EpiDerm culture dissociation, the labeling of the different cell types can take time depending on user expertise, delaying DNA-damage measurement, and allowing for DNA repair in the interim if cells are not properly maintained at 4 °C. To circumvent this problem, keratinocyte and fibroblast cell lines were created with stable integration of fluorophore sequences. Recognition software was then developed that could register cell-location based on the initial presence of a fluorophore prior to alkaline treatment and electrophoresis. This gave the system additional utility in that cells that express any generic fluorophore, e.g., GFP, mCherry etc., can have their position registered. The fluorophore is then destroyed by the lysis and alkylating steps in the comet assay, allowing a second round of DNA labeling using dyes; in our case, this was the comet SYBR Gold stain. HaCaT cells expressing mEmerald, a bright EGFP variant [28], and NHDF-expressing mCherry were loaded into a standard agarose CometChip. The chip was then imaged. Figure 4A shows the loading of the fluorescently labeled cells in a single macro-well of the 96-well chip. The CometChip was then processed by the standard protocol. In short, part of the methodology, namely the lysis and alkaline treatment, destroyed the fluorescent protein, allowing for secondary labeling of the immobilized cells. SYBR Gold was used to detect the comet DNA (Figure 4B). The pre- and post-comet images were then aligned, and the cells were assigned a numerical identifier. The previously described CometNet was then used to measure comet-centric parameters. Using this analytical approach allowed the system to measure the labeled cell-types in a single comet macro-well, greatly increasing both utility and reproducibility.

### 3.5. Characterization of 57-Compound NTP Library for Cytotoxicity and Genotoxicity in 2D Cultures

To test if the Immuno-CometChip platform was suitable for chemical-compound screening, we utilized an open-source chemical library of 57 compounds offered by the **NTP** through the National Institute of Environmental Health Sciences, USA. The library contains a broad spectrum of chemicals, including potential DNA damaging agents, procarcinogens/carcinogens, or common chemicals to which environment exposure might be relatively likely. Initially, the library was provided randomly and utilized in a single blind study. All stock concentrations were fixed at 20 mM in DMSO. We performed a preliminary screening using serial dilution in DMSO vehicle. We observed that concentrations of DMSO > 10% strongly reduced viability vs. control (Figure 5); thus, experiments with 2D HaCaT and normal human epidermal keratinocytes (**NHEK**) did not exceed 1% DMSO.

To classify the compounds as damaging agents, we determined their cytotoxicity and genotoxicity profiles. XTT cell viability assays were performed in 2D primary NHEK and HaCaT keratinocytes, as well as in HepG2 cells after exposure to the 57 NTP compounds for 48 h. While most of the compounds were found to be non-cytotoxic at 200 μM, five of the agents were cytotoxic in all three cell lines, as indicated by decreased cell viability compared to the 1% DMSO control (Figure 5A–C). Cytotoxic compounds that reduced viability by >50% in all three cell types tested include **B1, C6, C9, C10,** and **E6** (significance determined using post hoc analysis, including Dunnett’s multiple comparison test). Five compounds were cytotoxic only to HaCaT cells (**C7**, **C8**, **D11**, **E2** and **E8**), and three were cytotoxic to NHEK alone (**B9, E1,** and **E7)**. CometChip assays were then performed on HaCaT and NHEK keratinocytes following exposure to the same 57 NTP compounds. HaCaT cells were also exposed to compounds in the presence of FBS or pretreated with S9 human liver fractions. In total, 73% (8/11) of compounds that induced cytotoxicity at 200 µM in HaCaT cells at 24 h also induce genotoxicity after either 1 h of exposure at 1 mM, 200 µM at 3 h, or both, while some compounds induced genotoxicity but not cytotoxicity (Table 1). It is important to note that, whereas genotoxicity was tested at an early time point (1 or 3 h), cytotoxicity was examined later (48 h). Thus, it likely that compounds that were identified as both genotoxic and cytotoxic rapidly induced DNA damage, resulting in eventual cell death.

Interestingly, pretreatment with S9 reduced DNA damage induced by many of the compounds (Figure 5D, row 3, **“HaCaT + S9”)**, but it increased damage induced by others. Thus, it may be more relevant to examine compounds that are metabolized by endogenous skin-resident xenobiotic-metabolizing enzymes (rather than S9 liver fractions) to determine topical genotoxicity, as described further in Section 2.7.

### 3.6. Characterization of Select NTP Compounds for Cytotoxicity and Genotoxicity in 3D Cultures

Three-dimensional skin equivalents (full-thickness EpiDerm) were exposed to NTP compounds that were identified as cytotoxic, genotoxic, or both in the first NTP screen with 2D cells, and enzyme-dissociated cells were prepared and subjected to XTT cell viability assays. The comet assays were then used to determine cytotoxicity (Figure 6A, **histogram**) and genotoxicity (Figure 6A,B **dot plot**). A streamlined loading and washing protocol produced a DNA-damage signature for some of the NTP compounds. EpiDerm tested with chemicals that gave a DNA-damage response in 2D did not necessarily show a similar pattern of cytotoxicity and genotoxicity (Table 1). The 57 individual compound names, along with their corresponding cytotoxicity and genotoxicity profile, after decoding and validation are shown in Appendix A. Overall, the platform provided a promising method for the primary screening of the compound library.

### 3.7. CYP Luciferase Platform for Classification of Procarcinogens

We next asked if any these compounds, such as polyaromatic hydrocarbons, might be converted from pro-carcinogens to carcinogens by skin-resident cytochrome p450 by utilizing a CYP1A1 and CYP1B1 activity assay employing Luciferin-CEE luminescence to determine endogenous CYP 1A1/1B1 (CYP Glow, Promega). We focused on cytochrome p450 1A1, 1A2, and 1B1 (CYP1A1 or CYP1B1), which confer the majority of CYP activity present in human skin [29,30]. Cytochrome p450 oxidizes Luciferin-CEE to generate chemiluminescence. We were thus able to generate a screening platform to detect the cytochrome p450 activity in vitro.

The concentration range for benzo(a)pyrene (**BaP**)-induced CYP1A1 and CYP1A1 expression was first determined by RT-PCR and qPCR in HaCaT cells (Figure 7A,B). Additionally, CYP luciferase activity was induced by 10 µM BaP. Upregulation of CYP1A1/B1 by BaP was observed previously via binding of the AhR to the dioxin/xenobiotic response elements of the *CYP1A1/B1* gene promoter as a heterodimer with ARNT [31]. CYP1A1 activity assays were verified using two-fold dilutions of liver S9 fractions as a positive control (Figure 7C, **right**). Consistent with the results of HaCaT cells, human keratinocytes isolated from 3D EpiDerm samples treated with the indicated doses of **BaP** also exhibited CYP1A1 activity, as shown by CYP450 luciferase activity assays (Figure 7C, **left)**.

Screening the 57-compound NTP library (Figure 7D) for CYP activity revealed three NTP compounds (**B3**, **E5**, and **E6**) that strongly upregulated CYP1A1/1B1 activity in HaCaT keratinocytes. These NTP compounds were later identified to be BaP, 2-Chloroethyldiethylammonium chloride, and zinc dibutyldithiocarbamate, respectively. This was consistent with the CYP1A1 activation observed in keratinocytes from EpiDerm in response to exposure to **BaP** (Figure 7C, left). Furthermore, we have previously shown that zinc dibutyldithiocarbamate was one of the strongest activators of DNA damage [24], but its association with CYP1A1 upregulation has not been previously observed. Two compounds including **A8** and **C9** (Micheler’s ketone and doxorubicin) inhibited CYP1A1/B1 activity, as has been observed with other compounds, including synthetic chemopreventive organoselenium compounds, and vinylic and acetylenic PAHs, which were previously identified as inhibitors of CYP1A1 and 1B1. [32,33,34].

Given that 3-methylcholanthrene (**3-MC**) was reported to induce CYP1A1/1B1 activity in mouse and human skin [35], and in spontaneously immortalized HaCaT [36], EpiDerm skin equivalent cultures were treated with **3-MC** as a positive control. Four compounds upregulated CYP1A1/1B1 activity, i.e., B3 (BaP), D11, E2 (curcumin), and E5 (2-Chloroethyldiethylammonium chloride; Figure 8A).

Immunofluorescent staining of EpiDerm treated with NTP compounds confirmed the induction of CYPB1 by additional compounds, whereas CYP1A1 was constitutively expressed (Figure 8B). In agreement with studies of human skin [30], CYP1A1 expression was limited to the basal layer, with suprabasal expression of CYP1B1 observed in some cases, including by C6 (cadmium chloride) C7 (hydroquinone), C8 (sodium dichromate), and C9 (adriamycin; Figure 8B). CdCl2 and Na-dichromate are oxidizing agents that are associated with DNA damage, and the upregulation of CYP1B1 was somewhat unexpected; however, it may be due to an indirect activation of the AhR, or AhR-independent CYPB1 pathway, as observed in *Ahr*−/− mice [37]. Hydroquinone has also been demonstrated to induce DNA damage in cultured cells [38]. These results together highlight the importance of testing skin equivalents to understand the activity or inactivity of topical procarcinogens. Both C9 (doxorubicin; Figure 8B) and E2 (curcumin; not shown) also upregulated CYP1B1 in the basal cells of EpiDerm (not shown), as is consistent with previous findings that these two compounds activate the AhR, thus upregulating *CYP1A1/1B1* gene expression [39,40].

## 4. Discussion

To mimic human skin, a 3D EpiDerm culture was used to replace traditional animal experiments, which are incompatible with large-scale screening [41,42]. Here, we selected 14 compounds from the NTP library that significantly decreased cell viability and increased DNA damage in HaCaT and NHEK keratinocytes, and then we used the CometChip assays to determine cytotoxicity and genotoxicity when applied topically to the 3D full-thickness skin model. We found that nine compounds significantly reduced cell viability in comparison to the acetone control. On the other hand, for the collagen CometChip, we dissociated the full thickness skin culture and labeled it with integrin β1. By overlapping the SYBR and Qdot655 fluorescent signals, the latter of which we specifically labeled keratinocytes, we could further quantify keratinocyte DNA damage without interference from other cell types. In summary, the pipeline we established is able to identify some keratinocyte-specific genotoxins in the absence of a separate keratinocyte isolation step, which was described previously [22,43,44]. Further studies validated at multiple locations and additional compounds will allow us to establish the accuracy and reproducibility of the test compared to 2D cultures.

We optimized the CometChip assay by incorporating 0.1% collagen to mimic a dermal component of human skin and to selectively capture keratinocytes. This method enabled the CometChip to trap basal and stem cells by protein–protein interactions and obviate interference from certain DNA damage in tissues containing multiple cell types. The traditional CometChip captured and aligned the cells in microwells by gravity loading, as well as the size of the cells. However, multiple cell loading within single wells is an issue that can interfere with post-analysis. In addition, cell clusters dramatically decreased the loading efficiency. We therefore developed a protocol where the cells were immunofluorescently stained with specific surface marker initially and then loaded into the corresponding ligand-coated CometChip, which was collagen in our case. The strong interaction between collagen and integrin-β1-containing heterodimers retains keratinocytes loaded into the microwell. This method can be adapted to any tissue that contains multiple cell types.

After performing the alkaline electrophoresis, the Qdot nanodye was able to maintain the signal to indicate the integrin β1 and further to stain the DNA by SYBR Green/Gold. We designed a program that can recognize double-stained cells and measure the specific DNA damage in keratinocytes. The platform could be a new tool to investigate the DNA damage in the tissue level in a high-throughput format. However, some drawbacks need to be addressed in the future. First, the cellular Qdot labeling process may cause additional DNA damage or repair. It was therefore important to have a suitable control, such as untreated cells, and a positive control, which consisted of cells treated with known DNA-damaging agents, to calibrate the %DNA in tail and tail moments. Furthermore, the nonspecific Qdot stain may lead to difficulty in the post-analysis, thus requiring threshold adjustment. Moreover, the size of microwell was critical to trap certain cells and prevent cells from washing out. We attempted to address these challenges by differential fluorophore labeling and the registration of cells in the CometChip prior to alkaline electrophoresis to assign degrees of DNA damage to specific cell types.

For the cytochrome p450 detection platform, we observed that BaP induces CYP1A1 in HaCaT and HepG2 cells but is constitutively expressed in 3D skin equivalents, while CYP1B1 is induced in the latter. Recent studies have suggested that the organotypic 3D cultures could mimic responses of tissues, allowing for a more accurate assessment of CYP expression compared to a 2D culture [42,43]. Moreover, the concept of receptor–ligand interaction in CometChip can be utilized to capture the specific cell types, and allow single cell analysis. The platform can be a cell-sorting tool similar to flow cytometry, with the advantage that it requires fewer cells and provides a real-time study without stabilizing and culturing the cells in order to perform subsequent experiments.

A comparison between the three CAS revealed that Opencomet identified a high percentage of the comets present but overestimated DNA damage when the levels were low. In contrast, the Trevigen CAS was more sensitive and could detect and quantify low levels of DNA damage but was more problematic when DNA-damage levels were above 50%. CometNet was able to find and quantitate low levels of DNA damage and was also able to maintain linearity over a greater range.

Several labs recently collaborated by using organotypic 3D human epidermal skin models, including EpiDerm™ (MatTek, Ashland, MA, USA) and Phenion FT (Henkel, Düsseldorf, Germany), that were exposed to test chemicals, and dissociated cells subjected to a comet assay [22,43,44]. However, in an attempt to overcome the problem of multiple cell types, the recently published studies required the isolation of keratinocytes and fibroblasts prior to the comet assay, a rigorous protocol that introduces unwanted assay artifacts [44]. Secondly, while the second study by the same group employed solvent controls for background DNA damage [43], a later study [44] did not. Some of these studies showed a dose-dependent increase in DNA damage with increased concentrations of genotoxic agents. However, a high background of keratinocyte-differentiation-associated damage was originally observed, with unacceptably high levels of DNA damage and resultant comet tail DNA in 25% of unexposed cells, limiting sensitivity, accuracy, and adaptability for a high-capacity format that will be necessary for screening large numbers of potentially genotoxic compounds [22]. Third, the addition of aphidicolin, an inhibitor of DNA polymerases α, δ, and ε, was used to observe damage in the latter studies [43,44], thus inhibiting both replication and hence also replication-associated DNA-repair pathways [45], and partially negating the use of 3D cultures that have separate proliferating and differentiating compartments. Finally, the microwells were used in low-throughput format comet slides. In the current study, we showed that we can overcome these limitations by (1) utilizing an Immuno-CometChip, (2) developing new software to ensure simultaneous measurement of basal cell markers and DNA damage, (3) including internal controls for DNA damage, and (4) measuring levels of skin cytochrome p450 activity that can convert procarcinogens to carcinogens. Some aspects of the previous skin comet were improved by simultaneously isolating and visualizing DNA damage in basal cells by using integrin β1 as a marker and collagen-binding receptor when complexed with the α2 integrin subunit. We also enhanced the assay by adapting a gel with 96 macro-wells, each containing roughly 400 microwells in a grid to enhance reproducibility and capacity (~40K cells at once), as described previously [24]. We varied a number of parameters, including the microwell diameter and depth and the washing stringencies. Several controls were employed to monitor success, including the use of cryopreserved cells to measure against standardized levels of DNA damage, and monitor the purity of basal cell isolation by using biotin-labeled anti-β1 integrins and Qdot® 655 streptavidin conjugate, which consists of semi-conductive nanoparticles that fluoresce under blue light. Once deposited, they can be visualized following the completion of the assay, reducing analysis time and allowing for the visualization of results. Our modified automated software was used to quantify levels of DNA damage only in basal cells. We were able to begin to validate the final Immuno-CometChip assay in the presence of several household chemicals obtained from Tox21, which might need activation via CYP enzymes. In the presented analysis, we avoided going into detail about the potential molecular mechanisms of each of the tested compounds. Many of the compounds used in this research were also used in previous publications of ours, in particular, Sykora et al. [24]. In this report we focused on the technology and the utility of the assay on different biological materials.

To conclude, we developed the Immuno-CometChip and protocols that can be utilized to measure DNA damage at tissue level, providing a high throughput platform to isolate epidermal basal and stem cells from complex mixtures of cells in organotypic culture, and to simultaneously quantify DNA damage.

## Figures and Tables

**Figure 1 genes-14-00630-f001:**
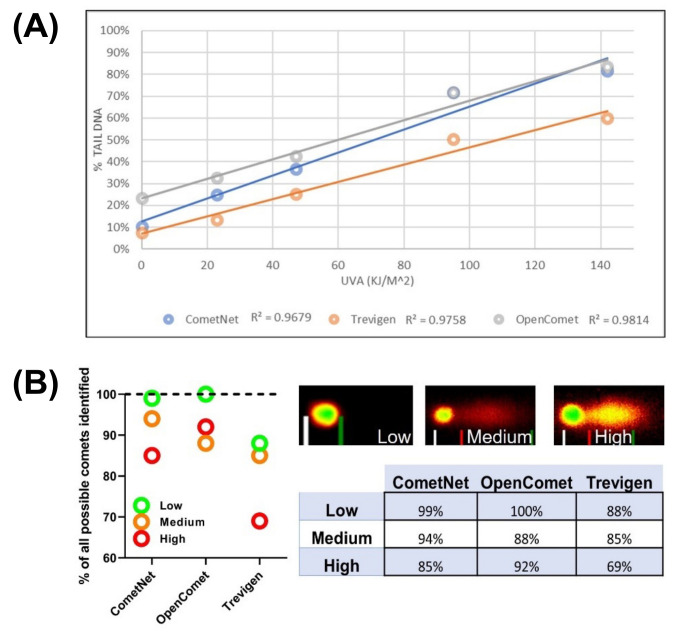
Comparison of the ability of three comet analysis (CAS) software packages to measure DNA damage over a linear range of DNA damage as measured by comet analysis. (**A**) Damage was induced using UVA (365 nm) light. Each data point represents averages of 100–300 comet cells from replicate biological experiments; r^2^ = linear regression of data set. (**B**) Both CometNet and OpenComet were able to identify significantly more comets than the Trevigen software. CometNet vs. Trevigen (*p* > 0.0001), OpenComet vs. Trevigen (*p* = 0.0001), *n* = 26 (technical sets). Comet images show representative comets of high, medium, and low levels of DNA damage.

**Figure 2 genes-14-00630-f002:**
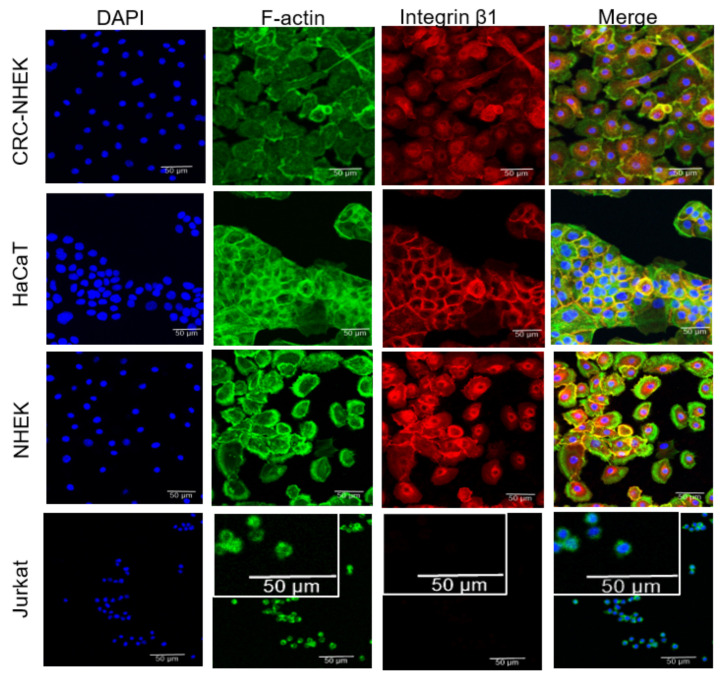
Human keratinocytes, CRC-HFK, HaCaT, and NHEK, but not Jurkat T cells, demonstrate strong integrin-β1 expression. Cells were stained with integrin-β1 antibody and coupled to Qdot-655. Nuclei were stained with DAPI. The cytoskeleton compartment was labeled with FITC-F-Actin. Jurkat cells were used as a negative control.

**Figure 3 genes-14-00630-f003:**
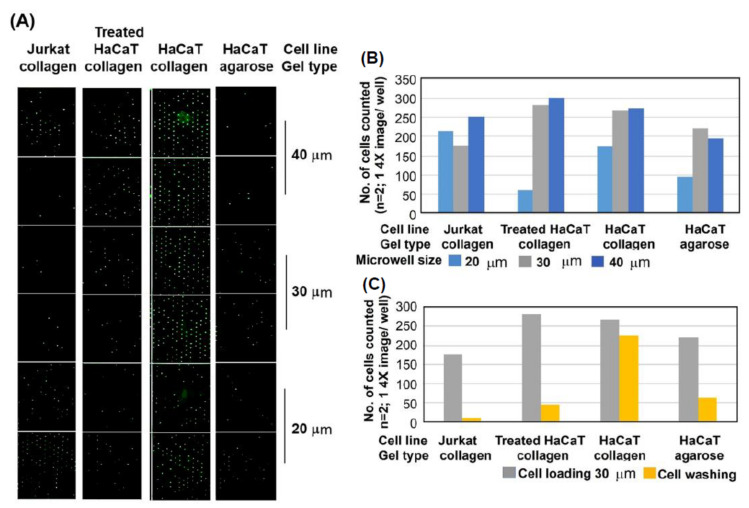
Loading study with 20–30–40 µm wells (A,B) and washing study using 30 µm pores (C). Jurkat cells are smaller and load well into 20, 30 and 40 µm pores, whereas HaCaT keratinocytes, expressing β1 integrin, are larger and load best into 30 and 40 µm wells. One wash is insufficient (**B**), but a second wash in 30 µm wells (**C**) shows that Jurkat cells do not bind to collagen I in the agarose microwells and are washed out, while HaCaT cells bind to collagen I in the microwells even after washing but are washed out if collagen I is omitted (“HaCaT agarose”) or if HaCaT cells are over-trypsinized to remove β1 integrin (“treated HaCaT collagen”), preventing binding to collagen I. (**C**) Corresponding CometChip graphs of (**A**).

**Figure 4 genes-14-00630-f004:**
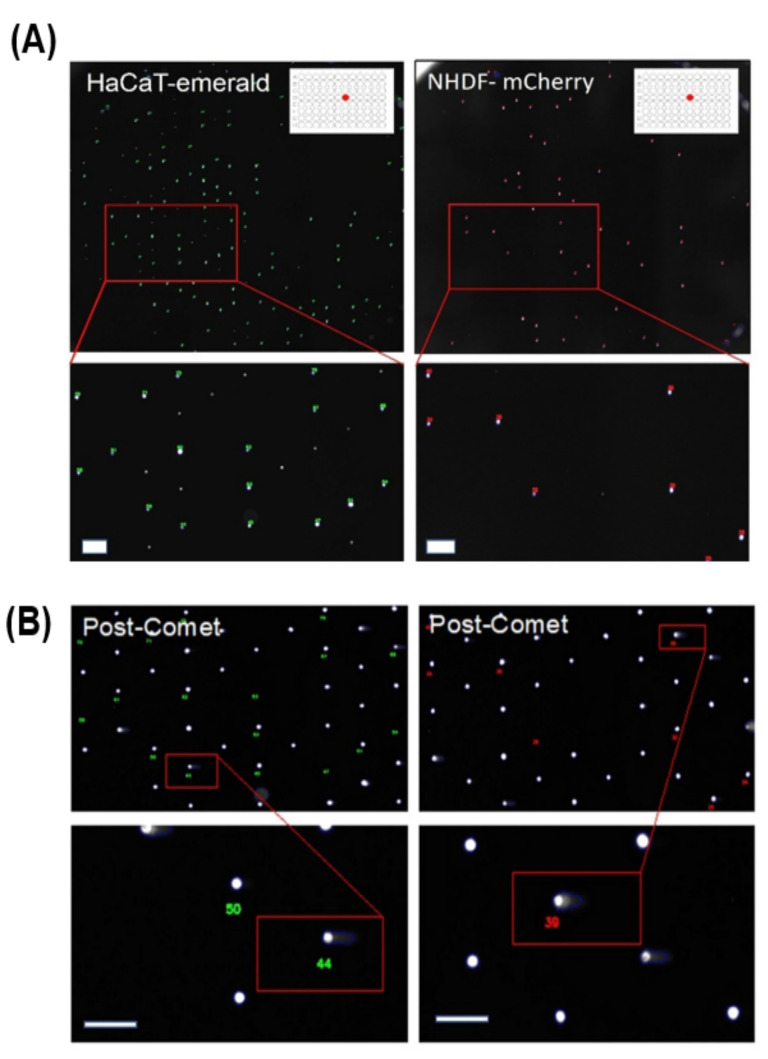
Representative images of dual-cell-type loaded well. (**A**) Pre-Comet mEmerald-labeled HaCaT and mCherry-labeled fibroblasts, were loaded into a single CometChip macro-well (upper images). Cell position was registered (lower images), and cells were identified using numbering before the CometChip was exposed to lysis and alkaline treatment. (**B**) Post-comet processing of CometChip shows the position of the differentially labeled fibroblasts and keratinocytes after comet protocol. In the image, fluorescence is SYBR Green DNA staining. The lower images show the registration of the cells and the blue outline around the cells shows the extent of the comet tail. Scale bar = 100 µm.

**Figure 5 genes-14-00630-f005:**
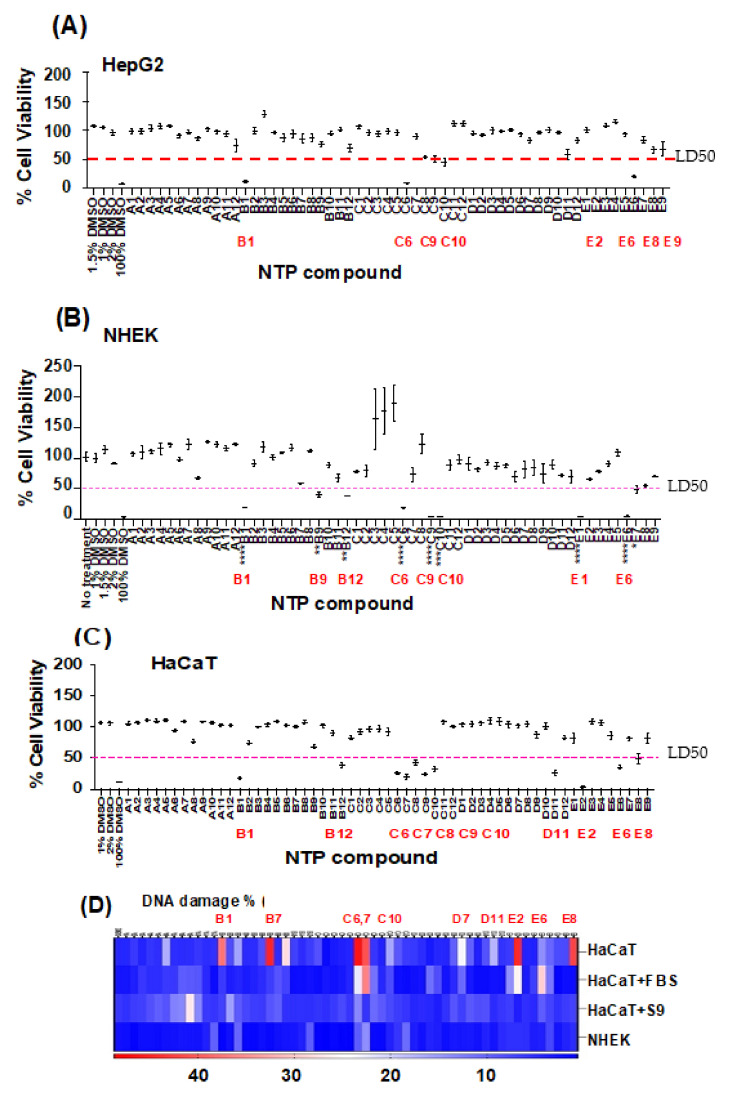
XTT cell viability assays of HepG2 (**A**) primary NHEK (**B**) and HaCaT keratinocytes (**C**) after treatment with 57 NTP compounds. HepG2 cells (**A**), primary NHEK keratinocytes (**B**), and HaCaT cells (**C**) were exposed to each of the 57 NTP compounds (designated A1–E10 (*x*-axis)) at 200 µM for 3 h and subjected to CometChip assay. (**D**) DNA damage (% tail DNA tail; *n* = 2) after exposure of HaCaT or NHEK cells with or without FBS or S9 activation. Error bars represent mean +/− SEM; one-way ANOVA in comparison to 1% DMSO control; *, **, *** and **** represent *p* < 0.05, *p* < 0.01, *p* < 0.001 and *p* < 0.0001, respectively.

**Figure 6 genes-14-00630-f006:**
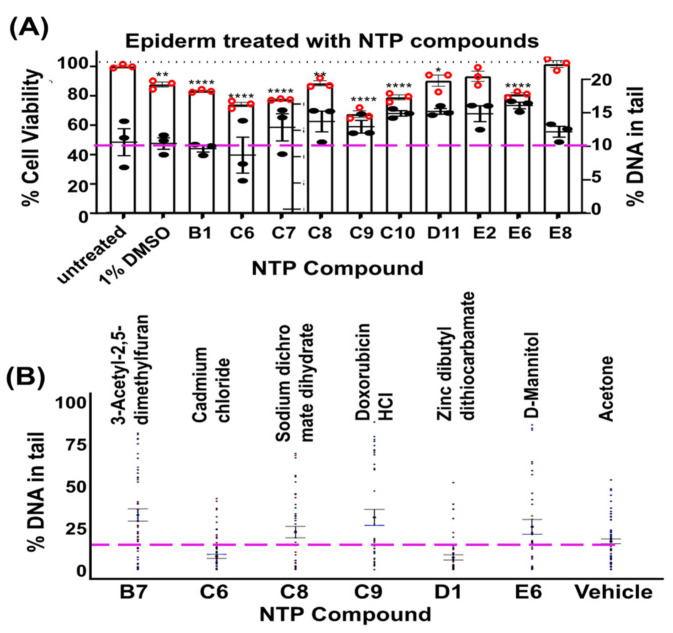
EpiDerm exposed to DNA-damaging agents. EpiDerm was topically exposed to each of the NTP compounds indicated. (**A**) Cell viability is shown as a histogram, with values on the left *y*-axis. % DNA in tail is overlaid as a 95% CI dot plot (*n* = 3), with values on the right *y*-axis, and a dashed line showing the value for vehicle control (10% DNA in tail). (**B**) Percent DNA in tail repeated for additional compounds, with dashed red line showing the value for vehicle control. *, **, and **** represent *p* values < 0.05, 0.01, and 0.0001, respectively.

**Figure 7 genes-14-00630-f007:**
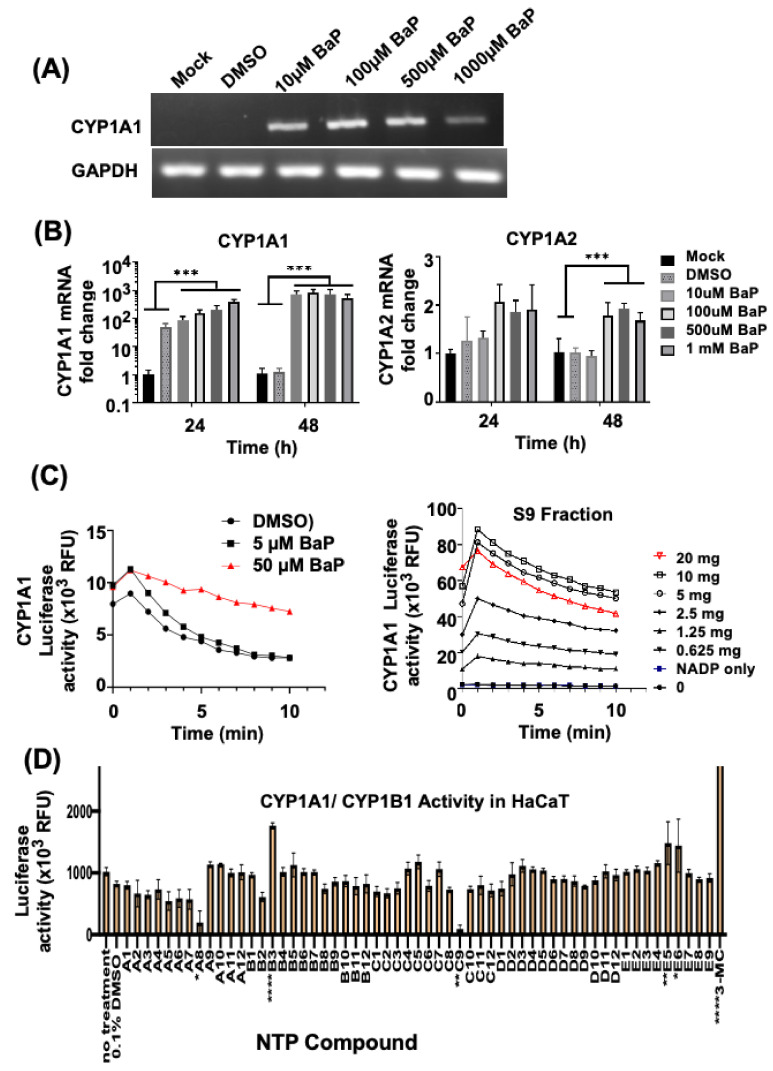
CYP activity in HaCaT keratinocytes after chemical treatment. Upregulation of CYP1A and CYP1A2 mRNA in HaCaT keratinocytes treated with Benzo[a]pyrene. (**A**) RNA (1 μg) was extracted at the indicated time points and subjected to reverse transcription to obtain cDNA, followed by RT-PCR, as well as (**B**) qPCR (SYBR Green) to detect CYP450 mRNA expression levels for CYP1A1 and CYP1A2. GAPDH served as an internal control. (**C**) CYP450 luciferase activity assay of human keratinocytes from EpiDerm samples treated with indicated doses of Benzopyrene (BaP) (left panel) and liver S9 fraction (right panel). Serial two-fold dilution of human S9 fraction was incubated with analog substrate (luciferin-CEE), and then the luciferase kinetic assay was performed and plotted (reading every 1 min). (**D**) CYP450 luciferase activity assay of HaCaT cells treated with each of the NTP compounds indicated. Data were collated from biological replicates. *, **, ***, and **** represent *p* values < 0.05, 0.01, 0.001, and 0.0001 respectively.

**Figure 8 genes-14-00630-f008:**
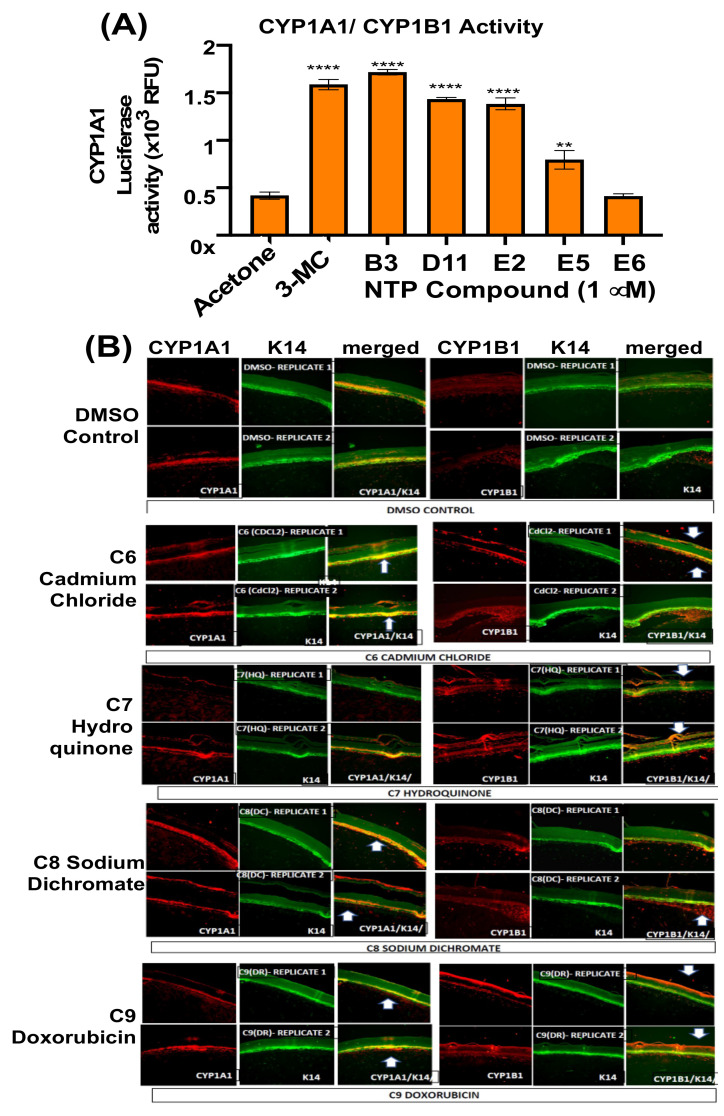
CYP activity (**A**) and expression (**B**) in EpiDerm skin equivalents after NTP compound treatment. EpiDerm was exposed to the indicated compounds, and 5 µm sections were derived and incubated using antibody specific for CYP1A1 or CYP1B1, followed by Alexa 488-conjugated or Alexa 594 conjugated secondary antibodies, as described in Materials and Methods. Equal exposures were captured by microscopy for a comparison of CYP expression levels. **, and **** represent *p* values < 0.01, and 0.0001, respectively.

**Table 1 genes-14-00630-t001:** Cytotoxicity and genotoxicity of compounds in 2D and 3D *.

	Cells/Media	Cytotoxicity (after 48 h)																										
		A1–12 **	B1	B2	B3	B7	B8	B9	B12	C6	C7	C8	C9	C10	C12	D1	D3	D6	D7	D8	D11	E1	E2	E5	E6	E7	E8	E9
1	HepG2																											
2	NHEK																											
3	HaCaT																											
	Genotoxicity																											
4	HaCaT 1 mM	A6																										
5	HaCaT 0.2 mM																											
6	HaCaT + FBS																											
7	HaCaT + S9	A7-A10																										
8	NHEK 0.2 mM	A12																										
9	EpiDerm									***																		

* Red boxes: cyto- or genotoxic. Gray boxes: negative for cyto- or genotoxicity. Black boxes: not tested. ** A1–12 shown in just one column, since most are non-toxic. *** Compound C6 was applied topically in acetone, but it is insoluble, and thus it did not show DNA damage.

## Data Availability

The data presented in this study are available on request from the corresponding authors. The data are not publicly available due to ongoing developmental IPs of the platforms.

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
