# Peer review of "Skin Immuno-CometChip in 3D vs. 2D Cultures to Screen Topical Toxins and Skin-Specific Cytochrome Inducers"

_genes, 2023, doi:10.3390/genes14030630_

Round 1
Reviewer 1 Report
An efficient method for detecting skin genotoxicity is necessary. Single-cell gel electrophoresis (SCGE) or the Comet has been used in a low throughput format until recently. In the manuscript titled “Novel Skin Immuno-CometChip in 3D vs. 2D Cultures to Screen Topical Toxins and Skin-Specific CYP Inducers”, Rosenthal et al show a method that can accurately predict genotoxicity using 3D skin cultures, as well as keratinocytes grown in 2D monolayers. Although some drawbacks need to be addressed such as additional DNA damage or repair in the cellular Qdot labeling process, this method is a new tool to investigate the DNA damage at the tissue level in a high throughput format. It might be a promising tool in the future. Minor concerns include:
1) It would be better to avoid “novel” and abbreviations in the title.
2) More molecular characterizations should be added after each treatment in this study.
Images should not be stretched.
Author Response
We thanks the reviewer for their positive comments and suggestions;
To directly address the reviewers three concerns we have edited the manuscript in the following ways;
1) It would be better to avoid “novel” and abbreviations in the title. We agree with the reviewer, and have removed the word "novel" from the title. We have changed the abbreviation, "CYP" to the term "cytochrome".
2) More molecular characterizations should be added after each treatment in this study. We thank the reviewer for the comment and concur that a molecular review on the treatments used in this study would increase the intellectual depth of the manuscript.
However, the main objective of this study was to show case the assay and the utility of the assay with a range of different biological materials. In this manuscript the chemicals used to induce response was a secondary element with much of the molecular description already covered in our previous publications (please refer to Sykora. et al. //doi.org/10.1038/s41598-018-20995-w) where a comprehensive review of the same compounds used in this study was undertaken. We have added a section in the discussion highlight this fact. The novel study of the impact of CYP activity on DNA damage in this manuscript, has had a considerable amount of the results and discussion section devoted to the molecular description.
Images should not be stretched.
We have gone through all the figures and rectified the problem specifically with Figure 4, Figure 5D, Figure 6 and Figure 8.
Reviewer 2 Report
The article entitled "Novel Skin Immuno-CometChip in 3D vs. 2D Cultures to 2 Screen Topical Toxins and Skin-Specific CYP Inducers" is written in well representable manner with real time impact of research for well being of mankind.
Author Response
We thank the reviewer for taking the time to read the manuscript.